# Dual-level Bias Mitigation via Fairness-guided Distribution Discrepancy

## Abstract

Modern artificial intelligence predominantly relies on pre-trained models, which are fine-tuned for specific downstream tasks rather than built from scratch. However, a key challenge persists: the fairness of learned representations in pre-trained models is not guaranteed when transferred to new tasks, potentially leading to biased outcomes, even if fairness constraints are applied during the original training. To address this issue, we propose Dual-level Bias Mitigation (DBM), which measures the fairness-guided distribution discrepancy between representations of different demographic groups. By optimizing both the fairness-guided distribution discrepancy and the task-specific objective, DBM ensures fairness at both the representation and task levels. Theoretically, we provide the generalization error bound of the fairness-guided distribution discrepancy to support the efficacy of our approach. Experimental results on multiple benchmark datasets demonstrate that DBM effectively mitigates bias in fine-tuned models on downstream tasks across a range of fairness metrics.

## 1 Introduction

Can we ensure fairness across various downstream tasks when fine-tuning a pre-trained model, without altering the original network architecture? In this paper, we aim to address this question by measuring the fairness-guided distribution discrepancy between the representations of different demographic groups to enforce fairness at both the representation and task levels.

Guaranteeing fairness in machine learning has become crucial as they are increasingly deployed in high-stake domains like healthcare (Chen et al., 2023), job recruitment (Faliagka et al., 2012) and credit approval (Khandani et al., 2010). Existing fairness approaches can be categorized into (1) pre-processing (Zemel et al., 2013; Calmon et al., 2017; Zhang et al., 2023), (2) in-processing (Bilal Zafar et al., 2016; Agarwal et al., 2018; Kamishima et al., 2012), and (3) post-processing (Hardt et al., 2016). Pre-processing and post-processing methods typically mitigate bias without modifying the model training process, where pre-processing removing bias from the data itself and employs standard machine learning methods for downstream tasks, and post-processing modifies the learned pre-trained model to achieve desirable fairness. In contrast, in-processing methods intervene during training by incorporating pre-defined fairness constraints, such as p%-rule (Biddle, 2005) and equalized odds (Hardt et al., 2016), in the objective function.

These methods often focus on creating fair models or representations for specific tasks, which limits their scalability in the era of big data and large models. Modern artificial intelligence increasingly relies on transfer learning, where pre-trained models are fine-tuned for specific tasks, rather than building models from scratch, particularly when dealing with large datasets for efficiency and effectiveness. This approach typically retains the internal representations learned by the pre-trained model while fine-tuning it for specific downstream tasks. To address fairness, recent works (Madras et al., 2018; Oneto et al., 2020) propose learning fair representations through neutralization or by leveraging inter-task similarities. However, focusing solely on representation-level fairness has limitations. Debiased representations can still leak sensitive information, as it is challenging to ensure complete removal of all sensitive information from the encoder. Moreover, enforcing strict fairness at the representation level may risk excluding task-relevant information (Du et al., 2021).

Motivated by the concept of mixing (Du et al., 2021; Chuang & Mroueh, 2021), which addresses bias through data augmentation and representation mixing, we propose a Dual-level Bias Mitiga-

tion (DBM) framework. DBM addresses bias at both the representation and task levels, providing a robust approach for mitigating bias in pre-trained models that are fine-tuned on biased data. It obtains representations of different demographic groups from a given pre-trained model and learns an in-processing module by minimizing the empirical risk over the set of mixed representations. Specifically, following the idea of R-divergence Zhao & Cao (2023), the fairness-guided distribution discrepancy between the two sets of representations is measured by the gap between the empirical risks of the pre-trained model and the in-processing module across the mixed representations. This fairness-guided distribution discrepancy is incorporated as a regularizer into the task-specific objective, aiming to minimize bias between the representations of different demographic groups and ensure fairness. By considering the task-specific objective while reducing the distribution discrepancy between the group representations, DBM achieves dual-level guarantees of fairness while ensuring accuracy.

## 2 PRELIMINARIES

In this section, we present our method for measuring task-specific fairness in debiased representations from the DNN head for downstream tasks. We define the probability distribution $\mathbb{P}$ on $\mathcal{X} \times \mathcal{S} \times \mathcal{Y}$, where $\mathcal{X} \in \mathbb{R}^d$ represents non-sensitive variables, $s = \{A, B\}$ is a binary sensitive variable, and $\mathcal{Y}$ is the binary classification output variable $\{-1, 1\}$. Specifically, $\mathbb{P}_A$ and $\mathbb{P}_B$ represent the distributions from samples from demographic groups $A$ and $B$, respectively. Accordingly, the training set $\mathbb{S} = \{\mathbf{x}_i, y_i, s_i\}_{i \in [N]}$ contains $N$ i.i.d. samples from $\mathbb{P}$, and $\mathbb{S}_A$ and $\mathbb{S}_B$ represent the datasets containing samples from demographic groups $A$ and $B$, respectively. We consider compositional models with a shared representation, expressed as $f(\mathbf{x}) = (g \circ h)(\mathbf{x})$, where $h : \mathcal{X} \to \mathcal{Z}$ is the representation learning function (i.e., DNN model head), and $g : \mathcal{Z} \to \mathcal{Y}$ is the task-specific classification function. The dimension of the internal representation is denoted by $r$, i.e., $\mathcal{Z} \subseteq \mathbb{R}^r$. To ensure fairness at the representation level, we require the conditional distribution of $h(\mathbf{x})$ to be identical across the two subgroups for every measurable subset $C \subset \mathbb{R}^r$:

$$P(h(\mathbf{x}) \in C \mid s = A) = P(h(\mathbf{x}) \in C \mid s = B). \tag{1}$$

Oneto et al. (2020) suggested that if demographic parity is satisfied at the representation level, then models built from such representations will also satisfy demographic parity. However, this condition is an ideal situation and does not always hold in practice. In fact, guaranteeing fairness at the representation level does not ensure fairness in final outcomes due to the complexity of downstream tasks. For example, task-specific transformations can introduce or amplify biases not present in the original representation (Madras et al., 2018). Therefore, it is crucial to evaluate and ensure fairness at the task-specific level, considering the entire pipeline from input to final output. To ensure fairness at the task-specific level, we require the conditional distribution of $g(h(\mathbf{x}))$ to be identical across the two subgroups for every class set $K \subseteq \mathcal{Y}$, given the fairness criteria of demographic parity:

$$P(f(\mathbf{x}) \in K \mid s = A) = P(f(\mathbf{x}) \in K \mid s = B). \tag{2}$$

While the constraint at the representation level can be achieved using various fair representation learning methods that remove sensitive information, the task-specific constraint is more challenging to handle.

When applying a representation learning function $h \in \mathcal{H}$ and a task-specific function $g \in \mathcal{G}$ to a distribution $\mathbb{P}$ and dataset $\mathbb{S}$, the corresponding expected risk and empirical risk are defined as:

$$\mathcal{E}_{\mathbb{P}}(f) = \mathbb{E}_{(\mathbf{x},y) \sim \mathbb{P}} \left[ \mathcal{L}(f(\mathbf{x}), y) \right], \quad \widehat{\mathcal{E}}_{\mathbb{S}}(f) = \frac{1}{|\mathbb{S}|} \sum_{(\mathbf{x},y) \in \mathbb{S}} \mathcal{L}(f(\mathbf{x}), y), \tag{3}$$

where $\mathcal{L}$ is a $L$-Lipschitz continuous loss function. The expected risk $\mathcal{E}_{\mathbb{P}}$ represents the theoretical performance over the true data distribution, while the empirical risk $\widehat{\mathcal{E}}_{\mathbb{S}}$ approximates this performance based on a finite dataset $\mathbb{S}$. The goal is to minimize both, with the empirical risk serving as a practical proxy for the expected risk. Ensuring that the gap between these two quantities remains small is critical for the model to generalize well from the training data to unseen samples. This balance becomes especially important when aiming for fairness across sensitive variables, as minimizing biased discrepancies across groups often requires careful consideration of both expected and empirical risks.

## 3 DUAL-LEVEL BIAS MITIGATION

In this section, we introduce DBM, a novel approach to mitigating bias in transfer learning scenarios. DBM focuses on two key levels of fairness: representation level and task-specific level. DBM measures and minimizes the fairness-guided distribution discrepancy between representations of different demographic groups. Specifically, the method begins by neutralizing the sensitive information through a mixing process of representations, and then incorporates a fairness-guided optimization that balances prediction errors across these groups. This dual-level approach ensures fairness both in the learned representations and in the downstream task, offering a robust framework for mitigating bias in pre-trained models fine-tuned on biased data.

**Representation Mixing.** The first level is at the representations. Following the same setting as in Du et al. (2021), after obtaining the representations from the pre-trained model, we randomly pair two representations from different demographic groups, say, $h(x_i \mid S = A)$ and $h(x_j \mid S = B)$ with the same value of $y$, and then neutralize them as $\frac{h(x_i|S=A)+h(x_j|S=B)}{2}$. This mixing process aims to obfuscate the sensitive information. To quantify the discrepancy between the representations of two demographic groups in the downstream task, i.e., the second level of fairness assurance, we introduce a fairness-guided distribution discrepancy measure.

**Fairness-guided Distribution Discrepancy.** Inspired by R-Divergence (Zhao & Cao, 2023), two distributions are likely identical if their optimal model yields the same expected risk. Accordingly, for a representation learning function $h$, the fairness-guided distribution discrepancy between $\mathbb{P}_A$ and $\mathbb{P}_B$ can be defined as:

$$\mathcal{D}(\mathbb{P}_A, \mathbb{P}_B | g^*, h) = \mathcal{E}_{\mathbb{P}_A}(g^* \circ h) - \mathcal{E}_{\mathbb{P}_B}(g^* \circ h), \tag{4}$$

where $g^*$ is the optimal model for the mixture distribution $\mathbb{U} = \frac{1}{2}\mathbb{P}_A + \frac{1}{2}\mathbb{P}_B$, i.e., $g^* \in \arg\min_{g \in \mathcal{G}} \mathcal{E}_{\mathbb{U}}(g \circ h)$. The fairness-guided distribution discrepancy can be estimated by the following estimator:

$$\widehat{\mathcal{D}}(\mathbb{S}_A, \mathbb{S}_B | \widehat{g}, h) = \widehat{\mathcal{E}}_{\mathbb{S}_A}(\widehat{g} \circ h) - \widehat{\mathcal{E}}_{\mathbb{S}_B}(\widehat{g} \circ h), \tag{5}$$

where $\widehat{g}$ is the minimizer for the mixed dataset $\mathbb{S}_A \cup \mathbb{S}_B$, i.e., $\widehat{g} \in \arg\min_{g \in \mathcal{G}} \widehat{\mathcal{E}}_{\mathbb{S}_A \cup \mathbb{S}_B}(g \circ h)$. Our method is based on regularized empirical risk minimization, combining a prediction error term and an estimated fairness-guided distribution discrepancy term. Specifically, we aim to solve the following optimization problem:

$$\min_{h \in \mathcal{H}, g \in \mathcal{G}} \frac{1}{N} \sum_{i=1}^{N} \mathcal{L}(g \circ h(x_i), y_i) + \alpha \widehat{\mathcal{D}}(\mathbb{S}_A, \mathbb{S}_B | \widehat{g}, h), \tag{6}$$

where $\alpha$ is a positive parameter that trades off between minimizing error and minimizing unfairness. The optimization in Eq. (6) is performed over classes $\mathcal{H}$ and $\mathcal{G}$ of possible representation and task specific functions, respectively. This formulation allows us to jointly optimize for task-specific performance and fairness, with the estimated fairness-guided distribution discrepancy serving as a measure of unfairness between the representations of different demographic groups.

**Implementation.** For the mixed representation part, we introduce an external module that takes the output of the pre-trained model, sensitive attributes, and downstream task information as input to process the representations for specific tasks. Based on the mixed representation, we then train the downstream task using fairness-guided distribution discrepancy. Compared to existing in-processing constraint methods, our proposed fairness-guided distribution discrepancy offers a comparable fairness guarantee with a much simpler implementation.

## 4 THEORETICAL GUARANTEES

In this section, we present generalization error bound of the fairness-guide distribution discrepancy. We have the following lemma for estimating the inherent difference between two datasets using a binary classifier.

**Definition 1.** *Let $\mathcal{G}$ be a hypothesis space with VC dimension $d$. For any $h \in \mathcal{H}$, considering the symmetric difference hypothesis space $\mathcal{G}\Delta\mathcal{G}$ which is the set of hypotheses for some $g, g' \in \mathcal{G}$:*

$$v \in \mathcal{G}\Delta\mathcal{G} \iff v(\mathbf{z}) = g(\mathbf{z}) \oplus g'(\mathbf{z}),$$

*where $\oplus$ is the XOR function. Therefore, every hypothesis $v \in \mathcal{G}\Delta\mathcal{G}$ is the set of disagreements between two hypotheses in $\mathcal{G}$. The empirical risk of a binary classifier which is learned for distinguishing between samples from $\mathbb{S}_A$ and $\mathbb{S}_B$ is defined as:*

$$\epsilon(\mathbb{S}_A, \mathbb{S}_B, \mathcal{G}) = \min_{v \in \mathcal{G}\Delta\mathcal{G}} \left[ \frac{1}{N} \sum_{\mathbf{x}:(v \circ h)(\mathbf{x})=0} \mathbf{I}\left[\mathbf{x} \in \mathbb{S}_A\right] + \frac{1}{N} \sum_{\mathbf{x}:(v \circ h)(\mathbf{x})=1} \mathbf{I}\left[\mathbf{x} \in \mathbb{S}_B\right] \right],$$

*where $\mathbf{I}\left[\mathbf{x} \in \mathbb{S}\right]$ is the binary indicator variable which is 1 when $\mathbf{x} \in \mathbb{S}$.*

Now, we derive a bound on the discrepancy between distributions of the two sensitive groups, formalized as the following theorem.

**Theorem 1.** *Let $\mathcal{D}$ denote the unbiased distribution discrepancy between the distributions $\mathbb{P}_A$ and $\mathbb{P}_B$, associated with the two sensitive groups. Similarly, let $\widehat{\mathcal{D}}$ represent the estimated unbiased distribution discrepancy between the datasets $\mathbb{S}_A \sim \mathbb{P}_A$ and $\mathbb{S}_B \sim \mathbb{P}_B$. Given a hypothesis $h \in \mathcal{H}$ where $|h(\mathbf{x})| \le B$ for $\mathbf{x} \in \mathcal{X}$, and considering the linear space $\mathcal{G} = \{\mathbf{z} \mapsto \langle \mathbf{w}, \mathbf{z} \rangle : \|\mathbf{w}\|_2 \le 1\}$, we define $g^* \in \arg\min_{g \in \mathcal{G}} \mathcal{E}_{\mathbb{P}_A \cup \mathbb{P}_B}(g \circ h)$ and $\widehat{g} \in \arg\min_{g \in \mathcal{G}} \widehat{\mathcal{E}}_{\mathbb{S}_A \cup \mathbb{S}_B}(g \circ h)$. With probability at least $1 - \delta$ over the sample draw $(\mathbb{S}_A, \mathbb{S}_B)$, the following holds:*

$$|\mathcal{D}(\mathbb{P}_A, \mathbb{P}_B | g^*, h) - \widehat{\mathcal{D}}(\mathbb{S}_A, \mathbb{S}_B | \widehat{g}, h)| \le 1 - \epsilon(\mathbb{S}_A, \mathbb{S}_B, \mathcal{G}) + \frac{\sqrt{d\ln(2N)} + 3\sqrt{\ln(16/\delta)} + 2LB}{N}.$$

The detailed proof of Theorem 1 is provided in Appendix A. The theorem provides a probabilistic bound on the difference between the true distribution discrepancy $\mathcal{D}(\mathbb{P}_A, \mathbb{P}_B | g^*, h)$ and the empirical distribution discrepancy $\widehat{\mathcal{D}}(\mathbb{S}_A, \mathbb{S}_B | \widehat{g}, h)$. This bound depends on several key factors. First, the empirical classification accuracy $\epsilon(\mathbb{S}_A, \mathbb{S}_B, \mathcal{G})$, which reflects the model ability to distinguish between the two groups, directly affects the discrepancy; the closer this value is to 1, the smaller the difference between the true and empirical distributions. Second, the VC dimension $d$, a measure of the complexity of the hypothesis space $\mathcal{G}$, influences the bound, with more complex spaces leading to larger potential gaps between the empirical and true discrepancies. Third, the sample size $N$ plays a crucial role, as larger datasets reduce the discrepancy by ensuring empirical estimates converge to expected values. Finally, the Lipschitz constant $L$ and representation bound $B$ help regulate the regularity of hypothesis space, preventing the loss function from growing too rapidly and keeping the learned representations within a reasonable range. Together, these factors guarantee that the difference between the true and empirical discrepancies remain small with high probability, providing reliable fairness measures during learning process.

## 5 EXPERIMENT

In the following sections, we first describe our experimental setup, including the datasets, baselines, and evaluation metrics. We then compare our proposed method against several related baselines and state-of-the-art techniques across multiple tasks, including both tabular and image tasks.

### 5.1 EXPERIMENT SETUP

**Architectures:** For the tabular datasets, we employ a Multi-Layer Perceptron (MLP) architecture for our pre-trained model. This MLP consists of two fully connected layers, each followed by a ReLU activation function. For the image dataset, we employ ResNet18 as the pre-trained model for feature extraction. The classification head is a two-hidden-layer MLP that takes the representations extracted by the pre-trained models and performs the final classification task.

**Baselines:** To evaluate the effectiveness and robustness of our proposed method, we compare our approach with several existing methods, including the fair representation learning methods described in Du et al. (2021) (**RNF**), the constraint on representations using Maximum Mean Discrepancy with Gaussian kernel, as proposed in Oneto et al. (2020) (**M_MMD**), the fairness constraints imposed on downstream tasks, specifically Equalized Odds (Donini et al., 2018) (**EO-FERM**), fair learning with Wasserstein distance (Jiang et al., 2020) (**W-FERM**), and robust fair empirical risk minimization (Baharlouei et al., 2024) ($f$-**FERM**).

**Datasets:** We evaluate the performance of our proposed method using three commonly employed benchmark datasets in related studies: income prediction (**Adult**), recidivism prediction (**COMPAS**), and two image datasets: Modified Labeled Faces in the Wild (**LFW+a**) and Celeb-Faces Attributes (**CelebA**). The Adult dataset comprises 30,717 records of individual annual incomes, aiming to predict if an individual earns over $50,000 annually, with gender as the sensitive attribute. The COMPAS dataset includes 5,554 instances predicting defendant recidivism, using race as the sensitive attribute. For image data, we utilize the modified Labeled Faces in the Wild Home (LFW+a) (Wolf et al., 2011) dataset, which we augment with attributes like gender and race. The task is to classify gender, with HeavyMakeup as the sensitive variable, given its strong correlation with female in previous research. The CelebA dataset is utilized to discern the label HeavyMakeup, considering gender as the sensitive variable where biases have been noted towards female.

**Evaluation Metrics:** We use the percentage of misclassifications (**ERR**) to measure the prediction performance. To measure fairness violations, we employ two metrics. The first is $\Delta_{\text{DP}} = |\mathbb{E}(\hat{Y} = Y \mid S = A) - \mathbb{E}(\hat{Y} = Y \mid S = B)|$, which quantifies the disparity in accuracy between demographic groups. The second is $\Delta_{\text{EO}} = |P(\hat{Y} = 1 \mid S = A, Y = y) - P(\hat{Y} = 1 \mid S = B, Y = y)|, \quad \forall y \in \{0, 1\}$, which measures the difference in true positive and false positive rates between groups. Lower values of $\Delta_{\text{DP}}$ and $\Delta_{\text{EO}}$ indicate smaller fairness violations.

**Experimental Settings:** Our experimental design consists of two main sets. The first compares our method with baselines across the three datasets. The second explores a scenario where observational labels are influenced by bias, simulated by flipping labels based on sensitive attributes and true labels. We test these methods with symmetrical bias levels of 20% and 40%. For tabular datasets, we conduct 10 runs with random splits, while for the two image datasets, we perform 3 runs, reporting mean results and standard deviations. All experiments are performed with GPU NVIDIA A30 with 86 GB memory.

## 5.2 MAIN RESULTS

As shown in Table 1, the proposed method demonstrates superior performance by consistently achieving low error rates while maintaining competitive or best fairness metrics across different datasets. RNF and $M_{\text{MMD}}$, which intervene at the representation level, generally perform worse than the task-specific methods and our proposed method. RNF consistently has higher ERR across all datasets compared to other methods. $M_{\text{MMD}}$ shows improvement over RNF but still falls short of the task-specific methods in most cases. EO-FERM, W-FERM, and $f$-FERM show varying performance across datasets. $f$-FERM performs well on the Adult dataset, achieving the second-best ERR after DBM. W-FERM shows strong performance on the LFW+a dataset, closely following DBM in ERR. EO-FERM performs consistently well across all datasets, often achieving a good balance between ERR and fairness metrics. DBM appears to achieve the best balance between accuracy (low ERR) and fairness (low $\Delta_{\text{DP}}$ and $\Delta_{\text{EO}}$) across all datasets. Other methods sometimes achieve better fairness metrics at the cost of higher error rates, or vice versa.

**Evaluation under Label Bias.** In this section, we present our second experimental setting, which evaluate scenarios where sensitive attributes influence the labels. We replicate the same experimental conditions on the LFW+a dataset, but introduce artificial bias by flipping labels with probabilities of 20% and 40% respectively. The results are presented in Table 2. From the results, we can observe that DBM still outperforms the other baselines under the settings of label bias, especially when the bias amount increases, which showcase our proposed one is more robust to the case in bias setting. The performance of other two task-level intervention methods of W-FERM and $f$-FERM drop a lot when the bias amount increase to 40%. It is worth noting that, for MMMD, though $\Delta_{\text{EO}} = 0$, it has the highest measure in $\Delta_{\text{DP}}$. This indicates that it has not achieved perfect fairness, but this might be because the prediction errors are large for both groups.

## 5.3 ABLATION STUDIES

As discussed in the introduction, the discrepancy measure is model-oriented. Therefore, we conduct ablation studies on various architectures by modifying the classification head of the pre-trained models. Specifically, we implement three MLP variants with one, three, and five hidden layers, respectively. Additionally, we evaluate different pre-trained models, including a CNN with three

Table 1: Evaluation of prediction errors and fairness violations across benchmark datasets. Methods that achieve the lowest prediction errors and fairness violations are highlighted using bold font.

| | | Representation | | Task level | | | Our |
|---|---|---|---|---|---|---|---|
| Dataset | Metric | RNF | $M_{MMD}$ | EO-FERM | W-FERM | $f$-FERM | DBM |
| Adult | ERR(%↓) | $21.91_{\pm0.59}$ | $18.35_{\pm1.48}$ | $16.87_{\pm0.35}$ | $22.30_{\pm3.62}$ | $15.71_{\pm0.40}$ | $\mathbf{15.29}_{\pm0.13}$ |
| | $\Delta_{DP}(\downarrow)$ | $0.16_{\pm0.02}$ | $0.13_{\pm0.01}$ | $0.12_{\pm0.01}$ | $0.16_{\pm0.06}$ | $0.12_{\pm0.01}$ | $\mathbf{0.11}_{\pm0.01}$ |
| | $\Delta_{EO}(\downarrow)$ | $0.05_{\pm0.06}$ | $\mathbf{0.04}_{\pm0.02}$ | $0.05_{\pm0.02}$ | $0.10_{\pm0.02}$ | $0.09_{\pm0.03}$ | $0.09_{\pm0.02}$ |
| Compas | ERR(%↓) | $49.97_{\pm0.71}$ | $31.69_{\pm1.33}$ | $36.25_{\pm0.08}$ | $32.22_{\pm1.58}$ | $33.10_{\pm0.97}$ | $\mathbf{31.04}_{\pm0.96}$ |
| | $\Delta_{DP}(\downarrow)$ | $0.08_{\pm0.03}$ | $\mathbf{0.01}_{\pm0.01}$ | $0.10_{\pm0.06}$ | $0.02_{\pm0.02}$ | $0.03_{\pm0.03}$ | $\mathbf{0.01}_{\pm0.01}$ |
| | $\Delta_{EO}(\downarrow)$ | $0.06_{\pm0.03}$ | $0.19_{\pm0.02}$ | $\mathbf{0.05}_{\pm0.01}$ | $0.20_{\pm0.04}$ | $0.18_{\pm0.03}$ | $0.10_{\pm0.19}$ |
| LFW+a | ERR(%↓) | $14.70_{\pm1.99}$ | $11.96_{\pm1.01}$ | $11.59_{\pm2.43}$ | $11.34_{\pm0.13}$ | $16.61_{\pm0.21}$ | $\mathbf{10.55}_{\pm1.95}$ |
| | $\Delta_{DP}(\downarrow)$ | $0.06_{\pm0.03}$ | $0.04_{\pm0.02}$ | $\mathbf{0.03}_{\pm0.02}$ | $0.06_{\pm0.03}$ | $0.05_{\pm0.04}$ | $\mathbf{0.03}_{\pm0.01}$ |
| | $\Delta_{EO}(\downarrow)$ | $0.02_{\pm0.01}$ | $0.03_{\pm0.01}$ | $0.03_{\pm0.02}$ | $0.04_{\pm0.02}$ | $\mathbf{0.01}_{\pm0.01}$ | $\mathbf{0.01}_{\pm0.01}$ |
| CelebA | ERR(%↓) | $17.16_{\pm0.56}$ | $33.70_{\pm0.65}$ | $15.11_{\pm0.47}$ | $16.81_{\pm0.92}$ | $15.52_{\pm0.62}$ | $\mathbf{14.54}_{\pm0.23}$ |
| | $\Delta_{DP}(\downarrow)$ | $0.29_{\pm0.03}$ | $0.64_{\pm0.01}$ | $0.28_{\pm0.08}$ | $0.29_{\pm0.10}$ | $0.28_{\pm0.11}$ | $\mathbf{0.26}_{\pm0.02}$ |
| | $\Delta_{EO}(\downarrow)$ | $0.75_{\pm0.18}$ | $\mathbf{0.33}_{\pm0.08}$ | $0.71_{\pm0.16}$ | $0.69_{\pm0.22}$ | $0.80_{\pm0.26}$ | $0.75_{\pm0.05}$ |

Table 2: Evaluation of accuracy and fairness violations on LFW+a dataset under the label bias setting with 20% and 40% bias amount.

| | 20% | | | 40% | | |
|---|---|---|---|---|---|---|
| Method | ERR(%↓) | $\Delta_{DP}(\downarrow)$ | $\Delta_{EO}(\downarrow)$ | ERR(%↓) | $\Delta_{DP}(\downarrow)$ | $\Delta_{EO}(\downarrow)$ |
| RNF | $16.09_{\pm0.31}$ | $0.04_{\pm0.03}$ | $0.01_{\pm0.01}$ | $18.31_{\pm1.67}$ | $0.04_{\pm0.02}$ | $0.02_{\pm0.01}$ |
| $M_{MMD}$ | $21.91_{\pm0.00}$ | $0.08_{\pm0.00}$ | $\mathbf{0.00}_{\pm0.00}$ | $22.78_{\pm0.00}$ | $0.10_{\pm0.00}$ | $\mathbf{0.00}_{\pm0.00}$ |
| EO-FERM | $14.21_{\pm0.58}$ | $\mathbf{0.03}_{\pm0.01}$ | $0.01_{\pm0.01}$ | $15.68_{\pm1.93}$ | $0.05_{\pm0.02}$ | $0.04_{\pm0.02}$ |
| W-FERM | $\mathbf{12.63}_{\pm0.35}$ | $0.04_{\pm0.01}$ | $0.05_{\pm0.01}$ | $20.21_{\pm0.00}$ | $0.05_{\pm0.00}$ | $0.00_{\pm0.00}$ |
| $f$-FERM | $14.80_{\pm1.07}$ | $\mathbf{0.03}_{\pm0.01}$ | $0.02_{\pm0.01}$ | $20.16_{\pm5.93}$ | $0.05_{\pm0.02}$ | $0.04_{\pm0.02}$ |
| DBM | $12.80_{\pm1.07}$ | $\mathbf{0.03}_{\pm0.00}$ | $0.02_{\pm0.00}$ | $\mathbf{14.16}_{\pm1.29}$ | $\mathbf{0.03}_{\pm0.02}$ | $0.02_{\pm0.02}$ |

convolutional layers and three max-pooling layers, as well as two zero-shot predictors, CLIP-RN50 and ViT-B/16 (Dosovitskiy et al., 2021). From the results shown in Table 3, we observe that modifying the classification head does not significantly affect accuracy or fairness violations. Similarly, changing the pre-trained model has a minimal impact on the results. We also compare the task-level methods (EO-FERM, W-FERM, $f$-FERM) with the proposed method on pre-trained model with fairness constraints (DP) in Fig. 1. For ERR, DBM achieves competitive performance, comparable to EO-FERM and W-FERM, but slightly lower than $f$-FERM. In terms of fairness metrics ($\Delta_{DP}$ and $\Delta_{EO}$), DBM demonstrates superior performance, achieving lower values than the baseline methods, indicating better fairness. Notably, DBM consistently stays below the fine-tuned performance on the fair pre-trained model, highlighting the robustness of our method in both accuracy and fairness.

## 5.4 HYPERPARAMETERS

We also conducted experimental analysis on hyperparameters. In the objective function, we have regularization term, we change the values of $\alpha$ from 0.1 to 1, and report the predictions errors and fairness violations under different settings of the hyperparameters. The plots on Fig. 2 shows that

Table 3: Evaluation of accuracy and fairness violations with different structures of classification head and pre-trained models on LFW+a dataset.

| Head | ERR(%↓) | $\Delta_{DP}(\downarrow)$ | $\Delta_{EO}(\downarrow)$ | Pre-trained | ERR(%↓) | $\Delta_{DP}(\downarrow)$ | $\Delta_{EO}(\downarrow)$ |
|---|---|---|---|---|---|---|---|
| MLP-1 | $10.43_{\pm0.99}$ | $0.03_{\pm0.01}$ | $0.01_{\pm0.01}$ | CNN | $11.03_{\pm0.93}$ | $0.04_{\pm0.01}$ | $0.03_{\pm0.02}$ |
| MLP-3 | $10.91_{\pm0.69}$ | $0.03_{\pm0.01}$ | $0.01_{\pm0.01}$ | CLIP-RN50 | $9.03_{\pm0.93}$ | $0.02_{\pm0.01}$ | $0.01_{\pm0.01}$ |
| MLP-5 | $10.99_{\pm0.57}$ | $0.04_{\pm0.02}$ | $0.02_{\pm0.01}$ | ViT-B/16 | $10.91_{\pm0.52}$ | $0.03_{\pm0.02}$ | $0.02_{\pm0.01}$ |

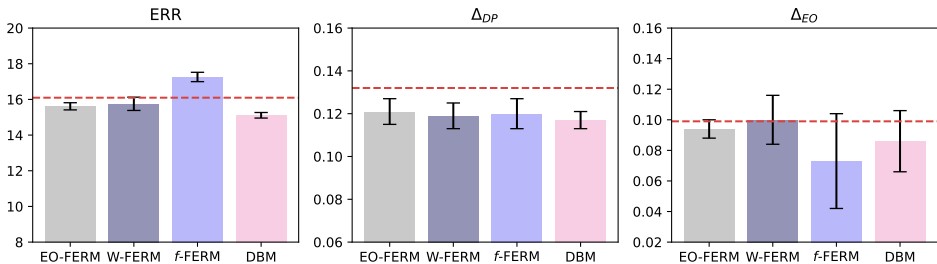

Figure 1: The performance was evaluated using a pre-trained model obtained with a fairness constraint (DP) on the Adult dataset. The red dashed horizontal line represents the results fine-tuned on the fair pre-trained model.

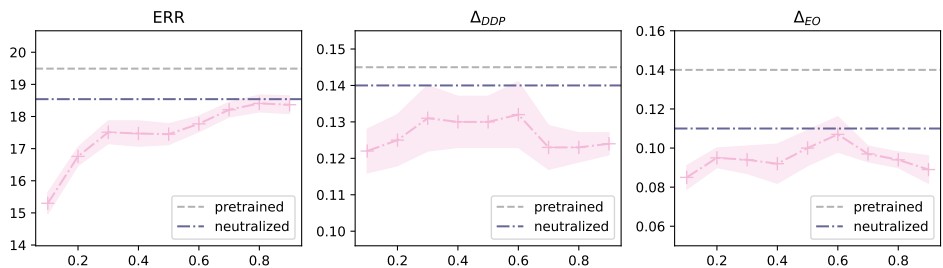

Figure 2: The performance under different value of $\alpha$ on Adult (results on other datasets are included in Appendix B). The gray dashed horizontal line represents the results obtained from pre-trained model; the blue dash-dot line represents the results obtained using neutralized representation.

the results of using neutralization representations obtained from pre-trained models with the classification head are better than just use the pre-trained model. As $\alpha$ increases with an appropriate value, the error rate steadily rises, while both fairness violations initially increase. However, when the intensity continue to increase, the fairness violations began to decrease while the error rate continue to increase. and then decrease. Despite these variations, the model performance generally remains below the constant levels of both using pre-trained model and using the neutralized representations for the fairness metrics, indicating improved fairness. However, for the error rate, the performance starts below but eventually rises between the pre-trained and neutralized levels as $\alpha$ increases, suggesting a trade-off between error rate and fairness improvements.

## 6 RELATED WORK

Ensuring fairness in the process from pre-trained models to downstream tasks can be approached from two perspectives. The first is to guarantee fairness at the level of representations learned by the pre-trained model. The second is to add different fairness constraints for different tasks after utilizing the pre-trained model. Many methods in this category aim to ensure fairness in downstream tasks by modifying the learned representations. Many methods aimed at ensuring fairness in downstream tasks focus on the representation level. For example, Du et al. (2021) adopted a method of neutralizing the representation, decorrelating its specificity towards certain groups. Cheng et al. (2021) apply contrasitive learning to debias, while another line of research uses adversarial learning to train debiased and transferable representations (Madras et al., 2018). These methods are similar to most representation learning approaches (Louizos et al., 2015; Zemel et al., 2013; Calmon et al., 2017; Lum & Johndrow, 2016; Zhao et al., 2020; Creager et al., 2019; Tan et al., 2020), which aim to ensure fairness in downstream tasks by guaranteeing the fairness of the representation.

Another category of methods adopts different learning strategies, such as using different in-processing methods for the downstream tasks to ensure fairness by applying different distribution measure. These methods typically follow an empirical risk minimization framework with fairness constraints to penalize the dependence between sensitive attributes and predictions. For example,

Baharlouei et al. (2024) applies f-divergence, Baharlouei et al. (2020) uses Rényi correlation, Lowy et al. (2022) employs $\chi^2$ divergence, Donini et al. (2018) utilizes $L_\infty$ distance, and Prost et al. (2019) implements Maximum Mean Discrepancy. The methods in this category can be either model-specific ((Bilal Zafar et al., 2015; 2016; Calders et al., 2009; Kamishima et al., 2012)) or generalizable ((Agarwal et al., 2018; Baharlouei et al., 2020; Lowy et al., 2022)). An alternative approach within this category leverages optimal transport learning (Gordaliza et al., 2019; Chiappa et al., 2020). Our method differs from these two categories. While ensuring the fairness of the representation, it is not truly possible to guarantee that the downstream task is definitely fair. On the other hand, adopting fairness in-processing for downstream tasks requires predefined fairness criteria. The advantage of our method is that it can consider the fairness of different tasks simultaneously without explicitly defining fairness.

## 7 CONCLUSION

In this paper, we introduced DBM, a novel approach to ensuring fairness in transfer learning scenarios using pre-trained models. DBM addresses the limitations of existing fairness methods by offering a dual-level fairness guarantee, tackling bias both at the representation and task-specific levels. By leveraging fairness-guided distribution discrepancy, our method effectively mitigates bias without altering the structure of the pre-trained model, making it highly adaptable and practical for real-world applications. DBM overcomes key challenges, such as the potential leakage of sensitive information from debiased representations and the risk of discarding task-relevant data when enforcing fairness. Through theoretical analysis and experimental evaluation on multiple benchmark datasets, we demonstrate the effectiveness of DBM in reducing bias across various fairness metrics.

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

## A  PROOF OF THEOREM 1

*Proof.* To prove the desired inequality, we begin by considering the difference between the true discrepancy and its empirical estimate $\left| \mathcal{D}(\mathbb{P}_A, \mathbb{P}_B \mid g^*, h) - \widehat{\mathcal{D}}(\mathbb{S}_A, \mathbb{S}_B \mid \widehat{g}, h) \right|$. Recall that the discrepancy between distributions $\mathbb{P}_A$ and $\mathbb{P}_B$ with respect to a hypothesis class $\mathcal{G}$ and a feature mapping $h$ is defined as:

$$\mathcal{D}(\mathbb{P}_A, \mathbb{P}_B \mid \mathcal{G}, h) = \sup_{g \in \mathcal{G}} |\mathcal{E}_{\mathbb{P}_A}(g \circ h) - \mathcal{E}_{\mathbb{P}_B}(g \circ h)|, \tag{7}$$

where $\mathcal{E}_{\mathbb{P}}(g \circ h)$ denotes the expected loss of the function $g \circ h$ under the distribution $\mathbb{P}$. Similarly, the empirical discrepancy is defined based on empirical samples $\mathbb{S}_A$ and $\mathbb{S}_B$. Our goal is to bound the

difference between the true discrepancy and its empirical estimate. Applying the triangle inequality twice, we have:

$$\left| \mathcal{D}(\mathbb{P}_A, \mathbb{P}_B \mid g^*, h) - \widehat{\mathcal{D}}(\mathbb{S}_A, \mathbb{S}_B \mid \widehat{g}, h) \right|$$

$$= \left| \left[ \mathcal{E}_{\mathbb{P}_A}(g^* \circ h) - \mathcal{E}_{\mathbb{P}_B}(g^* \circ h) \right] - \left[ \widehat{\mathcal{E}}_{\mathbb{S}_A}(\widehat{g} \circ h) - \widehat{\mathcal{E}}_{\mathbb{S}_B}(\widehat{g} \circ h) \right] \right|$$

$$= \left| \left[ \mathcal{E}_{\mathbb{P}_A}(g^* \circ h) - \widehat{\mathcal{E}}_{\mathbb{S}_A}(\widehat{g} \circ h) \right] - \left[ \mathcal{E}_{\mathbb{P}_B}(g^* \circ h) - \widehat{\mathcal{E}}_{\mathbb{S}_B}(\widehat{g} \circ h) \right] \right|$$

$$\leq \left| \mathcal{E}_{\mathbb{P}_A}(g^* \circ h) - \mathcal{E}_{\mathbb{P}_B}(g^* \circ h) \right| + \left| \mathcal{E}_{\mathbb{P}_A}(\widehat{g} \circ h) - \mathcal{E}_{\mathbb{P}_B}(\widehat{g} \circ h) \right|$$

$$+ \left| \mathcal{E}_{\mathbb{P}_A}(\widehat{g} \circ h) - \widehat{\mathcal{E}}_{\mathbb{S}_A}(\widehat{g} \circ h) \right| + \left| \mathcal{E}_{\mathbb{P}_B}(\widehat{g} \circ h) - \widehat{\mathcal{E}}_{\mathbb{S}_B}(\widehat{g} \circ h) \right|$$

$$= \underbrace{\left| \mathcal{E}_{\mathbb{P}_A}(g^* \circ h) - \mathcal{E}_{\mathbb{P}_B}(g^* \circ h) \right|}_{\mathcal{B}_1(g^*)} + \underbrace{\left| \mathcal{E}_{\mathbb{P}_A}(\widehat{g} \circ h) - \mathcal{E}_{\mathbb{P}_B}(\widehat{g} \circ h) \right|}_{\mathcal{B}_1(\widehat{g})}$$

$$+ \underbrace{\left| \mathcal{E}_{\mathbb{P}_A}(\widehat{g} \circ h) - \widehat{\mathcal{E}}_{\mathbb{S}_A}(\widehat{g} \circ h) \right|}_{\mathcal{B}_2} + \underbrace{\left| \mathcal{E}_{\mathbb{P}_B}(\widehat{g} \circ h) - \widehat{\mathcal{E}}_{\mathbb{S}_B}(\widehat{g} \circ h) \right|}_{\mathcal{B}_3}.$$

Next, we note that both occurrences of $\mathcal{B}_1$ involve the absolute difference of expectations over $\mathbb{P}_A$ and $\mathbb{P}_B$ for functions in $\mathcal{G}$. Since $g^*$ and $\widehat{g}$ are elements of $\mathcal{G}$, we can write:

$$\mathcal{B}_1(h) = \left| \mathcal{E}_{\mathbb{P}_A}(g \circ h) - \mathcal{E}_{\mathbb{P}_B}(g \circ h) \right|. \tag{8}$$

Therefore, combining the two $\mathcal{B}_1$ terms, we have:

$$2\mathcal{B}_1 = 2 \sup_{g \in \mathcal{G}} \left| \mathcal{E}_{\mathbb{P}_A}(g \circ h) - \mathcal{E}_{\mathbb{P}_B}(g \circ h) \right|. \tag{9}$$

Now, we proceed to bound each term.

**Bounding $\mathcal{B}_1$**  By definition of the discrepancy and using Lemma 1 and Lemma 2 from Ben-David et al. Shalev-Shwartz & Ben-David (2014), we have:

$$\mathcal{B}_1 = \sup_{g \in \mathcal{G}} \left| \mathcal{E}_{\mathbb{P}_A}(g \circ h) - \mathcal{E}_{\mathbb{P}_B}(g \circ h) \right|$$

$$= \max_{g \in \mathcal{G}} \left| \mathbb{E}_{\mathbf{x} \sim \mathbb{P}_A}[g(h(\mathbf{x}))] - \mathbb{E}_{\mathbf{x} \sim \mathbb{P}_B}[g(h(\mathbf{x}))] \right|$$

$$\leq 1 - \epsilon(\mathbb{S}_A, \mathbb{S}_B, \mathcal{G}) + 4\sqrt{\frac{d \ln(2N) + \ln(2/\delta)}{N}},$$

where $\epsilon(\mathbb{S}_A, \mathbb{S}_B, \mathcal{G})$ is the empirical estimate of the discrepancy, $d$ is the VC dimension of $\mathcal{G}$, $N$ is the number of samples, and $\delta$ is the confidence level.

**Bounding $\mathcal{B}_2$**  To bound $\mathcal{B}_2$, we utilize the Rademacher complexity $\mathcal{R}_N(\mathcal{G} \circ h)$ of the class $\mathcal{G} \circ h$ based on $N$ samples. First, recall that for any function $g \in \mathcal{G}$ and sample set $\mathbb{S}_A$, the deviation of the empirical mean from the true expectation can be bounded using the Rademacher complexity and concentration inequalities Awasthi et al. (2020):

$$\left| \mathcal{E}_{\mathbb{P}_A}(g \circ h) - \widehat{\mathcal{E}}_{\mathbb{S}_A}(g \circ h) \right| \leq 2\mathcal{R}_N(\mathcal{G} \circ h) + 3B\sqrt{\frac{\ln(2/\delta)}{2N}}, \tag{10}$$

where $B$ is an upper bound on the loss function, i.e., $|g(h(\mathbf{x}))| \leq B$. Using Talagrand's contraction lemma Mohri et al. (2018), if the loss function is Lipschitz continuous with Lipschitz constant $L$, we have:

$$\mathcal{R}_N(\mathcal{G} \circ h) \leq L\mathcal{R}_N(\mathcal{G} \circ h). \tag{11}$$

Assuming $\mathcal{R}_N(\mathcal{G} \circ h) \leq \frac{B}{\sqrt{N}}$, we get:

$$\mathcal{B}_2 \leq \frac{2LB}{\sqrt{N}} + 3B\sqrt{\frac{\ln(2/\delta)}{2N}}. \tag{12}$$

**Bounding $\mathcal{B}_3$**   Similarly, we can bound $\mathcal{B}_3$ using the same technique applied to $\mathbb{S}_B$:

$$\mathcal{B}_3 \leq \frac{2LB}{\sqrt{N}} + 3B\sqrt{\frac{\ln(2/\delta)}{2N}}. \tag{13}$$

**Combining the Bounds**   Substituting the bounds from equations equation 10, equation 10, and equation 13 back into inequality equation 8, we obtain:

$$\left| \mathcal{D}(\mathbb{P}_A, \mathbb{P}_B \mid g^*, h) - \widehat{\mathcal{D}}(\mathbb{S}_A, \mathbb{S}_B \mid \widehat{g}, h) \right| \leq 2 \left( 1 - \epsilon(\mathbb{S}_A, \mathbb{S}_B, \mathcal{G}) + 4\sqrt{\frac{d\ln(2N) + \ln(2/\delta)}{N}} \right)$$

$$+ 2 \left( \frac{2LB}{\sqrt{N}} + 3B\sqrt{\frac{\ln(2/\delta)}{2N}} \right)$$

$$= 2 \left( 1 - \epsilon(\mathbb{S}_A, \mathbb{S}_B, \mathcal{G}) \right) + 8\sqrt{\frac{d\ln(2N) + \ln(2/\delta)}{N}}$$

$$+ \frac{4LB}{\sqrt{N}} + 6B\sqrt{\frac{\ln(2/\delta)}{2N}}. \tag{14}$$

Simplifying and combining like terms, we get the final bound:

$$\left| \mathcal{D}(\mathbb{P}_A, \mathbb{P}_B \mid g^*, h) - \widehat{\mathcal{D}}(\mathbb{S}_A, \mathbb{S}_B \mid \widehat{g}, h) \right| \leq 2 \left( 1 - \epsilon(\mathbb{S}_A, \mathbb{S}_B, \mathcal{G}) \right) + C\sqrt{\frac{\ln(N/\delta)}{N}}, \tag{15}$$

where $C$ is a constant that depends on $d$, $L$, and $B$. This completes the proof. $\square$

# B   ADDITIONAL EXPERIMENTAL RESULTS

In this section, we include the results of performance error and fairness with varying $\alpha$ on other datasets. The plots in Fig. 3, which cover three additional datasets, show a consistent pattern: the results using neutralized representations obtained from pre-trained models with the classification head outperform those only fine-tuning the pre-trained model, as shown by the blue dash-dot line all being below the gray dashed line. As $\alpha$ increases, $\Delta_{DP}$ and $\Delta_{EO}$ both show a decreasing trend, particularly for the LFW+a and CelebA datasets, indicating that fairness improves when the value of $\alpha$ becomes larger. However, this trend is not well presented in the COMPAS dataset.

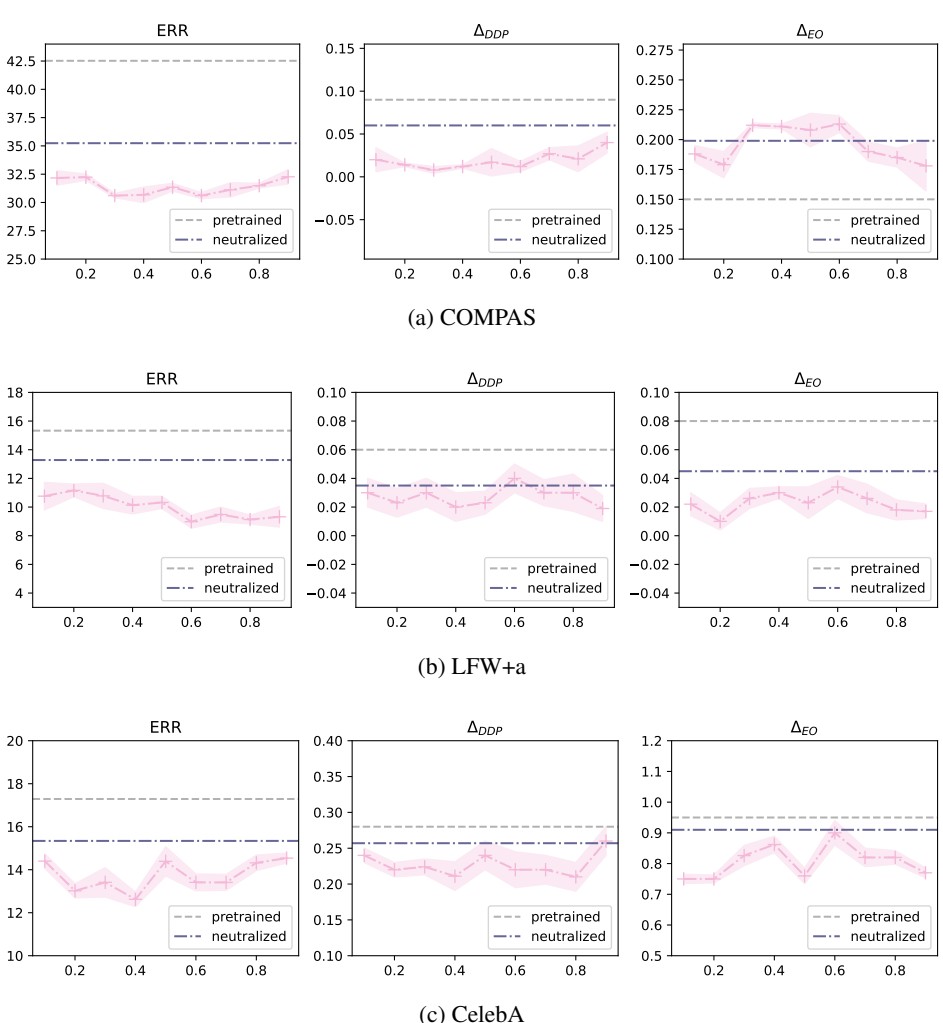

(a) COMPAS

(b) LFW+a

(c) CelebA

Figure 3: The performance under different value of $\alpha$ on the COMPAS, LFW+a and CelebA dataset. The gray dashed horizontal line represents the results obtained from pre-trained model; the blue dash-dot line represents the results obtained using neutralized representation.

