# OpenReview forum: "Dual-level Bias Mitigation via Fairness-guided Distribution Discrepancy"
_ICLR.cc/2025/Conference — Submitted to ICLR 2025_

### Official Review · Reviewer_Htoa · 2024-10-27

**Soundness:** 2
**Presentation:** 2
**Contribution:** 2
**Rating:** 3
**Confidence:** 4

**Summary:**

To address the fairness problem in transfer learning where the fairness of learned representations in pre-trained models is not guaranteed when transferred to new tasks, this paper proposed a method tackling bias both at the representation and task-specific levels. Specifically, the method employ representation mixing to obfuscate the sensitive information and a weighted objective  to trades off between minimizing error and minimizing unfairness. Generalization error bound of the fairness-guide distribution discrepancy is analyzed and empirical experiments are conducted to demonstrate the effectiveness of the proposed method.

**Strengths:**

1.	Compared with representation level baselines and task level baselines, experimental results on multiple benchmark datasets demonstrate the effectiveness across a range of fairness metrics.

**Weaknesses:**

1.	As stated in line 27, The key problem this paper studies is “Can we ensure fairness across various downstream tasks when fine-tuning a pre-trained model, without altering the original network architecture?” Although the sentence highlights “across various downstream tasks”, but the method studied in this paper is for training fair models for specific downstream tasks not generally for any downstream task. If it’s in this case, to my understanding, any existing in-processing method can be applied to finetuning pretrained models on downstream tasks? What makes this question special or challenging?
2.	Important experimental details are missing, making it difficult to understand and evaluate the reported experimental results. For example, how are the hyperparameters for baselines tuned? Particularly, baselines like f-FERM also involve a regularization term that balances the accuracy and fairness. What is the principle to choose the regularization weight and selected the results reported in Table 1? Do these pretrained model keep fixed or keep changing during finetuning on downstream tasks? How are the pretrained models trained, like Multi-Layer Perceptron for the tabular datasets? As this information is missing,  figure 1 is also difficult to understand.
3.	From section 3, the proposed method contains two operations. One is Representation Mixing, which is adopted from Du et al. (2021). Another is training with proposed Eqn 6, which is also studied in the literature like[1]. As the experimental settings are also unclear, the impact of the contribution is relatively limited.

[1 ] Technical Challenges for Training Fair Neural Networks

**Questions:**

1.	In line 92, it says “the task-specific constraint is more challenging to handle”.  This is confusing. To my knowledge, most existing in-processing methods are proposed for the task-specific constraints. Could the authors help clarify what specific challenges they are referring to regarding task-specific constraints, and how their approach addresses these challenges differently from existing in-processing methods?
2.	Although the paper emphasizes that the proposed method tackling bias both at the representation and task-specific levels, there is no result to support the fairness mitigation at the representation level.
3.	In line 234, what’s the purpose of the second experimental setting? Could the authors help explain the motivation behind this experimental setting and how it contributes to demonstrating the effectiveness or robustness of their method.
4.	In line 223, why do the authors take HeavyMakeup as sensitive variables and predict gender? This setting is unnormal. It makes more sense to predict HeavyMakeup and take gender as sensitive variables.

---

### Official Review · Reviewer_sK3Z · 2024-11-04

**Soundness:** 2
**Presentation:** 3
**Contribution:** 2
**Rating:** 5
**Confidence:** 4

**Summary:**

Instead of building models from scratch, most pre-trained models are fine-tuned for specific tasks. This approach can introduce fairness challenges, as the adapted representations may carry biases when transferred to new tasks.
To address this, the authors propose Dual-level Bias Mitigation (DBM), incorporating a novel approach to representation mixing that aims to reduce discrepancies in representations among datasets with sensitive attributes. At the task level, DBM introduces an additional regulation term to balance differences in predicted values across sensitive groups while maintaining prediction accuracy. This dual focus on representation and task levels ensures fairness across both dimensions.
Furthermore, the paper provides a theoretical generalization error bound for fairness-guided distribution discrepancy, supporting DBM’s effectiveness. Experimental results on various benchmark datasets validate DBM's ability to mitigate bias across diverse fairness metrics in fine-tuned models.

**Strengths:**

1. The paper is well-structured, clearly addressing both representation and task-specific fairness, with theoretical support provided separately. This clear organization makes the content easy to follow.
2. The proposed approach is straightforward yet effective: it mixes pre-trained representations of two sensitive attributes and trains downstream tasks with fairness-guided distribution discrepancy. This approach is practical to implement and shows promising results in the experiments.

**Weaknesses:**

1. The experimental section needs more explanation and a clearer conclusion to better highlight the findings.
2. The evaluation section should include more detail on $\delta_{DP}$ (Demographic Parity) and $\delta_{EO}$ (Equalized Odds) to clarify their relevance to fairness evaluation.

**Questions:**

1. In the experimental section, for the method you proposed in Tables 1 and 2, the $\delta_{EO}$ evaluation metric is not the lowest across three of the four datasets. Can you explain how this aligns with your claim that the difference in true positive and false positive rates between groups is low? (I understand that for the LFW+a dataset, the $M_{MMD}$ is 0.00, meaning it has no impact.)
2. Why did you choose to flip labels with probabilities of 20% and 40% in your experiments? You only conducted one test doubling the flipping rate and concluded that DBM outperforms other baselines under label bias, especially with increased bias. Can you validate this claim by using additional percentages?
3. In Table 3, I see you used different pre-trained models, but I couldn't find any comparative conclusions in your discussion. Could you explain the insights or conclusions you intended to draw from this?
4. Could you clarify your approach to mixing different representations and managing varying levels of bias? Specifically, is increasing bias achieved solely by flipping labels, or do you employ other methods? Additionally, how do you define and categorize different types of bias in your study?

---

### Official Review · Reviewer_6J9N · 2024-11-04

**Soundness:** 2
**Presentation:** 2
**Contribution:** 2
**Rating:** 3
**Confidence:** 4

**Summary:**

This paper proposes a Dual-level Bias Mitigation (DBM) that combines Representation Neutralization for Fairness (RNF) and R-Divergence to mitigate bias when fine-tuning pre-trained models for downstream tasks. Specifically, RNF is applied to enhance fairness at the representation level by neutralizing sensitive information within the encoder’s output. Additionally, R-Divergence is used as a regularization term during downstream task learning to reduce distributional discrepancies between groups with different sensitive attributes.

**Strengths:**

1. This paper tackles the significant and pressing challenges of model fairness and robustness, both of which are essential for the effective deployment of pre-trained models in real-world applications.

2. The proposed method incorporates well-grounded theoretical foundations, including RNF and R-divergence, which strengthen its approach to mitigate bias.

**Weaknesses:**

1. The novelty of this work is limited. The main contribution of this paper lies in its application of existing techniques, particularly R-Divergence, along with the presentation of a generalization error bound. However, R-Divergence is naively applied as a regularization term to measure discrepancies across groups with varying sensitive attributes.

2. A comparison with other recent fairness methods is needed. Although f-FERM (2024) is included as a recent baseline, further evaluation with additional up-to-date methods would strengthen the credibility of the experimental results.

3. Does the proposed method function in settings without annotations for sensitive attributes? Recently, many approaches, including RNF, have focused on addressing fairness (or robustness) without access to sensitive attribute labels, demonstrating strong performance. A comparison with existing methods in settings without sensitive attribute annotations would strengthen the proposed method. If such a comparison is not feasible, a discussion of this limitation is required.

4. The definitions of RNF and R-Divergence should be expanded to enhance clarity. In Sec. 3, a more detailed explanation of RNF is needed, including the role of neutralized labels in training and the hyperparameter lambda for controlling the degree of neutralization. Additionally, absolute value symbols appear to be missing on the right-hand side of equations (4) and (5).

**Questions:**

Please see the Weaknesses

---

### Meta-Review · Area_Chair_Y6gj · 2024-12-13

**Metareview:**

I have read all the materials of this paper including the manuscript, appendix, comments, and response. Based on collected information from all reviewers and my personal judgment, I can make the recommendation on this paper, reject. No objection from reviewers who participated in the internal discussion was raised against the reject recommendation.

**Research Questions**

The paper studies the fairness learning problem.

**Challenge Analysis**

The authors argue that the representation of pre-trained models is not guaranteed when transferred to new tasks, even if fairness constraints are applied in the pre-trained model.

**Philosophy**

I did not find any philosophy to tackle the above challenge. Instead, the authors directly provide their solution.

**Technique**

The authors propose dual-level bias mitigation by optimizing fairness-guided distribution discrepancy and task-specific objectives. However, Eq. (6) is a vanilla learning loss with an extra fairness constraint, which is quite standard in the fairness learning area. Moreover, it is unclear why task-specific objectives can mitigate unfair issues, especially the targeted challenges.

**Theoretical Analysis**

I don’t believe that every paper requires a theoretical analysis, especially if the analysis is disconnected from the proposed method—it may even detract from the paper's overall value.

**Experiments**

1.	The experimental setting does not simulate the pre-trained model for new tasks for tabular datasets. Moreover, the authors need a baseline where fairness constraints are applied in the pre-trained model.
2.	Fairness learning has caught extensive attention in the community. The authors need to provide rationality on the selection of baseline methods.
3.	Fairness learning is a dual-direction evaluation problem. It is not recommended to apply tables for demonstrating experimental results. Instead, the fairness and utility curves are preferred.

**Additional Comments On Reviewer Discussion:**

No objection from reviewers who participated in the internal discussion was raised against the reject recommendation.

---

### Decision · Program_Chairs · 2025-01-22

Reject